# Theia: Distilling Diverse Vision Foundation Models for Robot Learning

**Jinghuan Shang[1,2], Karl Schmeckpeper[1], Brandon B. May[1], Maria Vittoria Minniti[1], Tarik Kelestemur[1], David Watkins[1], Laura Herlant[1]**

[1]The AI Institute   [2]Stony Brook University
{jshang, kschmeckpeper, bmay, mminniti, tkelestemur, dwatkins, lherlant}@theaiinstitute.com

**Abstract:** Vision-based robot policy learning, which maps visual inputs to actions, necessitates a holistic understanding of diverse visual tasks beyond single-task needs like classification or segmentation. Inspired by this, we introduce Theia, a vision foundation model for robot learning that distills multiple off-the-shelf vision foundation models trained on varied vision tasks. Theia's rich visual representations encode diverse visual knowledge, enhancing downstream robot learning. Extensive experiments demonstrate that Theia outperforms its teacher models and prior robot learning models using less training data and smaller model sizes. Additionally, we quantify the quality of pre-trained visual representations and hypothesize that higher entropy in feature norm distributions leads to improved robot learning performance. Code, models, and demo are available here.

**Keywords:** Visual representation, Robot learning, Distillation, Foundation model

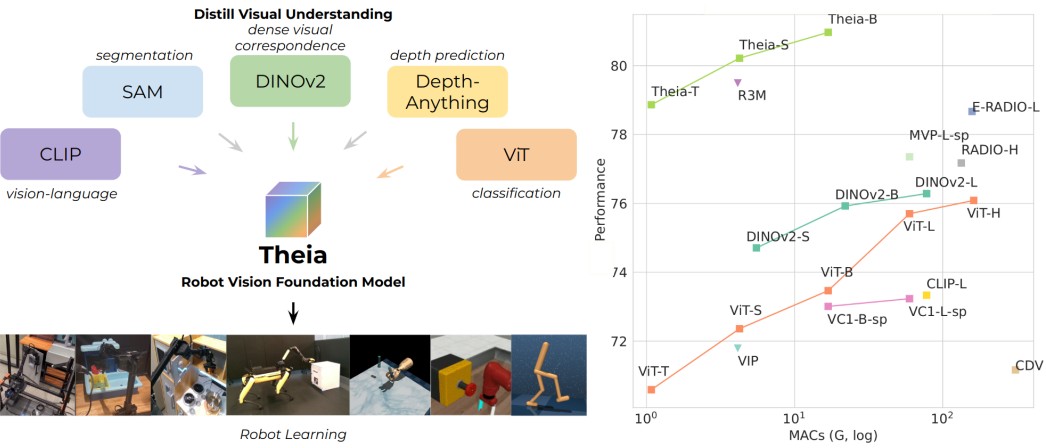

Figure 1: We introduce Theia, a model that distills multiple vision foundation models (VFMs) to provide better representations for robot learning (left). Theia achieves superior performance on robot learning tasks with less computation compared to standard VFMs and pre-trained models for robotics (right). Results shown are from the MuJoCo subset of tasks in CortexBench.

## 1 Introduction

Visual understanding, i.e., the process of abstracting high-dimensional visual signals like images and videos, includes many different sub-problems, from depth prediction [1] and vision-language correspondence [2, 3], to tasks ranging from coarse to fine granularity such as classification [4, 5] and object grounding [6, 7, 8], as well as tasks defined along spatial and temporal axes like segmentation [9, 10] and tracking [11]. Given this diversity, a long-standing effort in the vision community has been to develop models tailored to one or a few specific types of visual understanding tasks. In recent years, several models [4, 5, 10, 2, 3, 12, 13, 6, 14, 15, 10, 7, 16] have achieved remarkable generalizability to unseen domains and new tasks, and are commonly referred to as vision foundation models (VFMs).

8th Conference on Robot Learning (CoRL 2024), Munich, Germany.

Vision-based robot policy learning, which learns action policies from visual inputs, requires strong and diverse visual comprehension. These policies involve many implicit vision tasks such as object recognition and semantic grounding, where off-the-shelf VFMs corresponding to some well-defined tasks can be easily found, but there is no single model for all of vision tasks. Studies have shown that off-the-shelf VFMs such as CLIP [2] usually under-perform relative to visual representation models tailored for specific tasks in robot learning [17, 18, 19, 20, 21]. This fact reveals a gap between the needs of robot learning and the limited visual understanding capabilities of any individual VFM. Our motivation is different than all prior works on learning visual representation models for robotics, where those works focused primarily on improving training data [17, 21, 20], designing objective functions [17, 18], and directly taking advances of vision architectures [19]. We uniquely focus on improving the visual representation from the angle of solving multiple implicit visual understanding tasks, which will benefit the downstream robot learning.

In this work, we propose combining multiple large VFMs into a single, smaller model for robot learning that leverages diverse visual understanding abilities from VFMs. We achieve this via knowledge distillation [22]. Unlike conventional distillation from a larger model to a smaller model on the same task, we distill VFMs tailored for varied vision tasks to improve visual representation for robot learning, which is an *unseen* task for VFMs.

We introduce Theia, a robot vision foundation model that simultaneously distills off-the-shelf VFMs such as CLIP [2], DINOv2 [7], and ViT [5]. Compared to off-the-shelf VFMs [2, 5, 7] and prior works [19, 20], Theia offers both better pre-trained visual representations for higher downstream robot learning performance and reduced computational costs. Furthermore, training Theia only requires ImageNet [23] and about 150 GPU hours, in contrast to prior works which necessitate substantially more compute [19, 20, 17, 18]. To understand what makes a good visual representation for robot learning, we observe multiple factors that relate to the performance of downstream robot learning tasks. We hypothesize that higher entropy in representation norms [24] correlates with improved robot learning performance.

In summary, our contributions are:

- We introduce Theia, a model that combines knowledge from multiple VFMs into a single, smaller model using knowledge distillation with low training cost.
- Through extensive simulated and real-world experiments, we confirm that Theia's visual representations lead to better downstream robot learning with improved computational efficiency.
- We identify key factors relevant to robot learning performance, such as model size, the use of spatial tokens, and the entropy of representation norms, offering valuable insights for guiding future research on optimizing visual representations for robot learning.

## 2  Related Work

### 2.1  Visual Representations for Robot Learning

Visual representations are important for vision-based robot policies to parse high-dimensional visual signals. Visual representation learning can happen at different stages, including pre-training [17, 18, 20, 21], joint-learning with robot tasks [25, 26, 27, 28], or a combination of both using either trainable or frozen visual representations [29, 30]. Off-the-shelf visual encoders [4, 5, 2, 31, 32] can also provide visual representations for robot learning. Additionally, an important factor in training visual representations is the choice of data. ImageNet [23], as suggested by Dasari et al. [21], is a particularly effective pre-training dataset, while video datasets [33, 34] are also widely used. Training objectives and auxiliary tasks for visual representation learning vary, including data augmentation [25, 26], prediction tasks [28], contrastive learning [27, 35, 36], and self-supervised learning [19, 20]. Specifically, to handle invariance and equivariance in visual observations, inductive biases [37, 38] and constraints [39, 40] can be introduced into neural networks to improve visual representation quality. Unlike prior works, we build a robot vision foundation model from the perspective of merging the visual understanding abilities of VFMs via distillation. Concurrent work OpenVLA [32] uses pre-trained SigLIP [41] and DINOv2 [7] encoders, which suggests the benefit of diverse visual understanding in robot tasks.

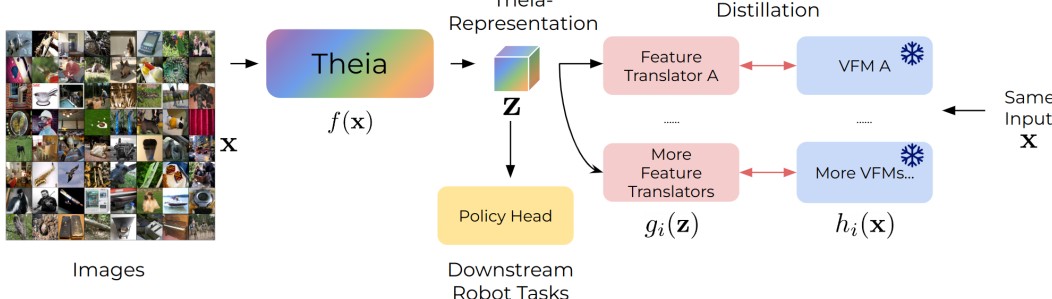

Figure 2: Theia distills multiple VFM features into one rich representation for robot learning. The feature translators $g_i(\mathbf{z})$ are supervised by the features from pretrained VFMs $h_i(\mathbf{x})$ during training time, then the distilled representation $\mathbf{z}$ is used as input to the policy head for robot learning tasks.

## 2.2 Vision Foundation Models

Vision foundation models trained on large-scale data exhibit strong task-specific performance and transferability to unseen domains and new tasks. VFMs can focus on single tasks, multiple tasks, or remain task-agnostic at the pre-training stage. For example, ViT [5], DeiT [42], and ConvNeXt [43] are designed for image classification, SAM [10] for semantic segmentation, and Depth-Anything [1] for depth prediction. When combined with Large Language Models (LLMs) [44, 3, 45, 46], these models can solve multiple visual tasks including referring segmentation, visual question answering (VQA), and image editing. Task-agnostic models like the DINO series [6, 7] are trained through self-distillation, while CLIP [2] is trained by aligning image-text pairs. Given the strong generalization capability of VFMs, robot learning should also benefit from leveraging the latent representations of pre-trained VFMs. which is the primary motivation for our research.

## 2.3 Knowledge Distillation in Vision Models

Knowledge distillation [22] compresses the knowledge from one or more larger models into a single smaller model. To leverage VFMs, which are usually computationally intensive, several studies have explored distilling them into more compact models. For example, some works distill a single VFM like SAM [10] into smaller variants [47, 48, 49]. There are also relevant works on combining two models: SAM-CLIP [50] merges SAM [10] and CLIP [2] into one model, while ViLD [51] and GroundingDINO [52] combine vision and language models to perform object detection. RA-DIO [53] is an agglomerative model that simultaneously distills CLIP, SAM, and DINOv2. These studies show that combined models can improve performance on downstream tasks or enable new applications. Similarly, in this work, we investigate whether combining VFMs will benefit robot learning. Notably, RADIO [53] is the closest approach to ours. The key differences between Theia and RADIO [53] are that (1) We aim to use our representation for robot learning tasks that none of the VFMs have covered in their pretraining tasks, (2) we distill only spatial tokens rather than both spatial and `[CLS]` tokens, (3) we choose a different set of teacher models, and (4) we show how each teacher model contributes to robot learning performance.

## 3  Method

**Overview.**  We introduce Theia, a framework that distills the knowledge of multiple VFMs into a smaller model, producing rich spatial representations for downstream vision-based robot learning. Figure 2 shows the overall design of Theia. Our model comprises a visual encoder (backbone) and a set of feature translators for distillation. Note that only the visual encoder is used to produce latent representations for downstream robot learning tasks.

**Architecture.**  Given an input image $\mathbf{x}$, the visual encoder $f(\cdot)$ produces a rich representation $\mathbf{z} = f(\mathbf{x})$ (called the Theia-representation), which is utilized for downstream robot learning tasks. We focus on backbone models that are smaller than typical VFMs, specifically using ViT-Tiny, Small, and Base [5, 42][1] motivated by the limited computing resources on robotic systems. We use Theia-Tiny (Theia-T), Theia-Small (Theia-S) and Theia-Base (Theia-B) to refer to Theia models

---

[1]For simplicity, we denote DeiT-Tiny and DeiT-Small [42] by ViT-Tiny and ViT-Small. DeiT was introduced by Touvron et al. [42] and the original ViT [5] does not have Tiny and Small models.

using a vision transformer backbone with the corresponding size. To train the Theia-representation, we perform feature distillation with the help of feature translators, which will be described below.

## 3.1 Rich Spatial Representation

The Theia-representation is a set of encoded tokens corresponding to input image patches. We choose spatial tokens because spatially-dense representations are the foundation for diverse visual understanding, as evidenced by the powerful per-patch features in DINOv2 [7]. Therefore, we aim to distill all spatial tokens and leave the `[CLS]` token untouched.

**Feature Translators.** Our goal is to supervise Theia-representations with teacher representations from various VFMs. We extract teacher representations $h_i(\mathbf{x})$ of VFMs at the last layer for CLIP [2], ViT [5] and DINOv2 [7], or before the decoders for SAM [10] and Depth-Anything [1]. Since a single representation cannot be learned to match all the teacher representations directly, feature translators $g_i(\cdot)$ are used to map the Theia-representation, $\mathbf{z}$, to each teacher representation. Feature translators are shallow CNNs to ensure that knowledge is distilled primarily into Theia's visual encoder. Details are available in the Appendix.

## 3.2 Training

**Distillation Objective.** Our training objective is matching the outputs of the feature translators with their corresponding teacher VFM representations. To achieve this, we use a combination of cosine and smooth-L1 losses [53] to match each pair of predicted and ground truth representations for the same image, taking their weighted average. Formally, our loss is

$$\mathcal{L}(\mathbf{x}; \theta) = \sum_i^M \alpha_i \big( \beta \mathcal{L}_{cos}(g_i(f(\mathbf{x})), h_i(\mathbf{x})) + (1 - \beta) \mathcal{L}_{smooth-L1}(g_i(f(\mathbf{x})), h_i(\mathbf{x})) \big), \quad (1)$$

where $\mathbf{x}$ is the input image, $M$ is the number of teacher VFMs, $\alpha_i$ is the loss weight for each teacher, and $\beta$ is the weight for balancing cosine loss and smooth-L1 loss respectively. In general, we set $\alpha_i = 1/M$ such that the loss weights each teacher equally. We empirically set $\beta = 0.9$ [53].

**Feature Normalization.** To properly accommodate the different scales of teacher representations, we first perform a normalization step. This helps us scale the loss of different teacher features evenly and avoid biasing (collapsing) to a teacher model with extremely larger norms. We perform the normalization on the teacher representations over each latent dimension, where mean and variance are calculated from all ImageNet training samples. Details are in Appendix A.

**Dataset.** We train our model on the ImageNet [23] training set for 50 epochs. We opt to use ImageNet because of its greater diversity compared to human videos [33, 54, 55, 56] and robot datasets [57, 58, 59, 60, 61] within the same number of images. This diversity has been experimentally shown to improve visual representation learning [21].

# 4 Experiments

## 4.1 Benchmark and Settings

To evaluate pre-trained visual representations, we use simulation tasks in CortexBench [20], which combines MuJoCo tasks (Adroit [62, 63], DeepMind Control Suite (DMC) [64], and Meta-World [65]), Habitat [66, 67] tasks (ImageNav [68], ObjectNav [69], and MobilePick [70].), and Trifinger [20] tasks[2]. ImageNav and MobilePick are reinforcement learning (RL) tasks and others are imitation learning (IL) tasks. We follow the experiment settings of Majumdar et al. [20] and report aggregated scores for rewards (DMC tasks) and success rates (all other tasks). For DMC tasks, raw rewards are divided by 10 to be in a scale consistent with the success rate. We also conduct real robot experiments, introduced in Section 4.3. We use the same policy heads for the each type of representations (vector or spatial tokens). Full experimental details are available in the Appendix.

---

[2]Due to reproducibility issues, we are not able to evaluate Move Cube, ObjectNav, and MobilePick tasks

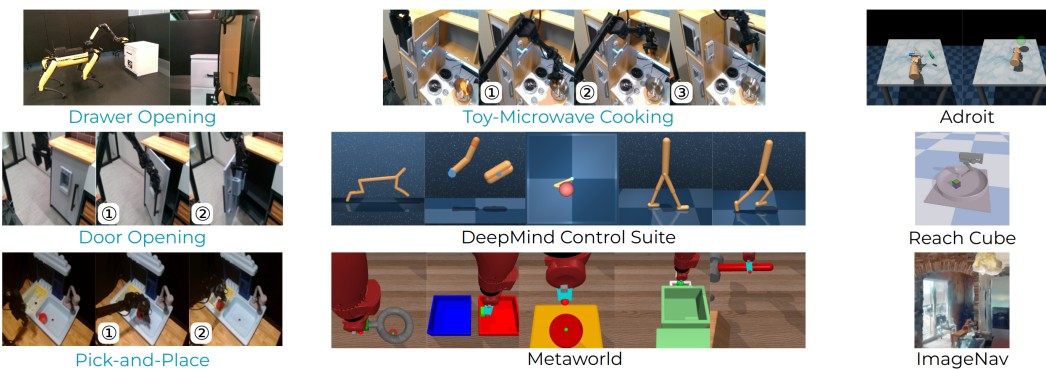

Figure 3: Simulation and real-world (labeled in blue) tasks used in this work. For simulated environments we show one image per task. For real-world tasks, we show images of key steps throughout the task labeled by numbers. A third-person view image shows the setup in Drawer Opening.

## 4.2 Simulation Results

We comprehensively evaluate Theia and baseline pre-trained models on the MuJoCo subset of CortexBench [20] to provide an overall assessment of pre-trained visual representations. We consider prior works on visual representations for robot learning, including R3M [17], VIP [18], MVP [19], and VC-1 [20], as well as agglomerative models for vision tasks RADIO and E-RADIO [53], and off-the-shelf vision foundation models ViT [5], DINOv2 [7], and CLIP [2]. We also test a naive concatination of three VFMs (CLIP, ViT, and DINOv2) which are used to train Theia, referred as CDV. All pre-trained representations are frozen in this experiment. Throughout this section, we answer the following questions:

- How does Theia perform compared to baselines?
- Which is more effective for visual representations: `[CLS]` or spatial tokens?
- How does robot learning performance scale with the size of the visual encoder?

**Theia Performance.** As shown in Figure 4, Theia outperforms all evaluated models, surpassing the performance of the best prior models, R3M and MVP, as well as agglomerative models for vision tasks RADIO and E-RADIO. We also tested a naive approach of using multiple VFMs (CLIP, DINOv2, and ViT) simultaneously by concatenating their spatial tokens channel-wise (CDV in Figure 4), but this performed much worse than using just the individual VFMs. Theia models scale effectively from tiny to base sizes, with Theia-S and Theia-B being the only models to break scores of 80 on this subset of CortexBench, even though they use only a small fraction of the inference computation required by comparable models. Theia's training is very efficient, using only the 1.2M images in ImageNet with a training time of about 150 GPU hours on NVIDIA H100 GPUs, compared to approximately 5M images used in prior works [17, 18, 19, 20] and 1B images used by RADIO [53].

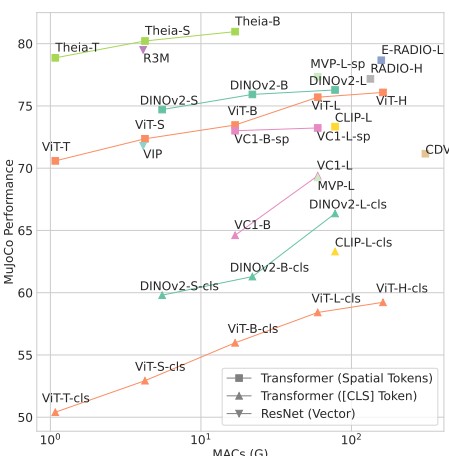

Figure 4: Performance on MuJoCo tasks vs. inference computation. Theia achieves the best performance with much less compute (MACs (G): Multiply-Accumulate operations in billions, log-scale).

**Spatial Tokens vs. `[CLS]` Token.** We evaluate Transformer-based models (all models in Figure 4 except R3M and VIP) using either their `[CLS]` token or spatial tokens [5] for downstream robot learning. To accommodate spatial tokens in the robot policy, we introduce extra shallow CNN layers at the input of the policy network, known as "compression layer" [71, 20]. Figure 4 shows the results of all models we evaluated, clearly showing that for Transformer-based models, providing spatial tokens is consistently better than using the `[CLS]` token for robot learning. This finding applies to both off-the-shelf VFMs and prior pre-trained representations, including MVP and VC-1.

Table 1: Mean CortexBench score across 14 tasks (2 in Adroit, 5 in MetaWorld, 5 in DMC, 1 in Trifinger (Reach Cube), and 1 in Habitat (ImageNav)).

| Model | **Theia-B (Ours)** | VC-1-L-sp [20] | MVP-L-sp [19] | R3M [17] | VIP [18] | E-RADIO [53] |
|---|---|---|---|---|---|---|
| Score | **79.79 ± 0.14** | 69.56 ± 0.80 | 77.42 ± 3.14 | 76.51 ± 0.79 | 71.18 ± 0.34 | 73.50 ± 1.69 |

Table 2: Real robot behavioral cloning results measured by success rate. Door Opening and Toy-Microwave Cooking are evaluated for 50 trials, and the others are evaluated for 20 trials. We report results for policies trained with either frozen (❄) or fine-tuned (🔥) visual encoders. For tasks having key stages (see Figure 3), we measure the success rates of achieving each stage separately.

| Model | # Parameters (M) | Door Opening ❄ | | Pick-and-Place ❄ | | Toy-Microwave Cooking 🔥 | | | Drawer Opening |
|---|---|---|---|---|---|---|---|---|---|
| | | ① Open | ② Fully Open | ① Pick | ② Place | ① Pick | ② Place | ③ Close the Door | ❄ / 🔥 |
| **Theia-B (Ours)** | 86 | **92%** | **66%** | **85%** | **75%** | **58%** | 52% | 40% | **85% / 100%** |
| E-RADIO [53] | 390 | 72% | 46% | 75% | 55% | **58%** | **54%** | **42%** | 15% / 80% |
| MVP-L-sp [19] | 303 | 32% | 2% | 55% | 35% | 40% | 26% | 18% | 30% / 65% |
| VC-1-L-sp [20] | 303 | 12% | 4% | 55% | 45% | 14% | 4% | 4% | 0% / 45% |
| R3M [17] | 24 | 48% | 36% | 35% | 10% | 36% | 26% | 18% | 0% / 55% |
| DINOv2-L [7] | 303 | 46% | 12% | 10% | 0% | 20% | 10% | 2% | 35% / 20% |
| ViT-H [5] | 632 | 18% | 4% | 15% | 0% | 52% | 44% | 40% | 45% / 85% |

**Scaling with Model Size.** We observe that most models, including Theia, achieve better performance with larger sizes. The scaling effect is more obvious when using the `[CLS]` token, probably because the `[CLS]` token encodes less information, making the size of the feature vector more critical. Different models that use spatial tokens also scale at varying rates. VC-1 shows only minor improvements when scaling from base to large, while ViT shows much larger improvements when scaling from tiny to huge model.

**CortexBench Results.** We compare Theia-B against baselines over the CortexBench evaluation suite [2]. The results in Table 1 confirm that Theia-B outperforms all other models.

## 4.3 Real World Robot Learning

Based on simulation performance, we test Theia-B and the best-performing baseline models: MVP-L [19], R3M [17], VC-1-L [20], DINOv2-L [7], ViT-H [5], and E-RADIO-L [53] for evaluation on real-world tasks. We employ four tasks (Figure 3): *Door Opening*, *Pick-and-Place*, and *Toy-Microwave Cooking* with a WidowX 250s arm, and *Drawer Opening* with a Boston Dynamics Spot. We train behavioral cloning policies on top of visual representations using conventional policy networks composed of CNNs and MLPs in the WidowX setup and diffusion policy [72] in the Spot setup. During testing, we vary the robot position for Door Opening and Drawer Opening, randomize the object position for Pick-and-Place, and randomize both the object positions and object types in Toy-Microwave Cooking. Full experimental settings, including details about the number of collected demonstrations, the interface used to collect them, and the policy architecture, are available in the Appendix (Sec. D.3).

Table 2 shows the success rates on these real-world tasks. Theia-B achieves the highest success rate across all tasks except Toy-Microwave Cooking. The results also highlight that the Theia-representation is useful for both conventional and diffusion-based policy heads, and for either freezing or fine-tuning the visual representation. E-RADIO is the most competitive model in this setting amongst all models compared, likely due to its similar distillation of VFMs and much larger training dataset. VC-1 has high task variance, performing poorly on Door Opening but adequately on Pick-and-Place. ViT-H works much better when being fine-tuned but DINOv2 does not, which could be caused by some fundamental differences in VFMs.

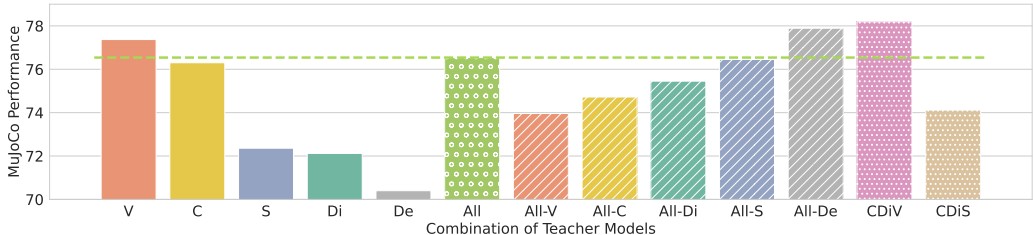

Figure 5: MuJoCo subset performance with respect to different combinations of teacher models to train Theia-T. Abbreviations of teacher models: **V**=ViT-H, **C**=CLIP-L, **S**=SAM-H, **Di**=DINOv2-L, **De**=Depth-Anything-L, **All**=all of five models (**CDeDiSV**), and **All−X**=taking **X** out of **All**.

Table 3: Ablation studies on model design. All experiments are based on Theia-T unless otherwise labeled. Scores are the average performance from the MuJoCo subset of CortexBench.

(a) Distillation losses

| MSE | Cos + L1 |
|---|---|
| 78.2 ± 1.0 | **78.9 ± 0.8** |

(b) Number of register tokens (Theia-B)

| 0 | 1 | 4 | 8 |
|---|---|---|---|
| 79.2 ± 1.2 | **80.3 ± 0.7** | 79.4 ± 0.8 | 79.6 ± 0.9 |

(c) Tokens to Distill. SP=Spatial Tokens

| | Distill SP | Distill [CLS] + SP |
|---|---|---|
| Use [CLS] | — | 50.0 ± 0.4 |
| Use SP | **78.9 ± 0.8** | 77.3 ± 2.4 |

**Selection of Teacher Models.** Theia is motivated by distilling many vision foundation models, but not all the models can be effectively integrated into one model nor do they contribute equally to downstream robot learning. In these experiments, we empirically identify the most effective combination of teacher models among five candidate VFMs: CLIP (referred to as **C**) [2], Depth-Anything (**De**) [1], DINOv2 (**Di**) [7], SAM (**S**) [10], and ViT (**V**) [5]. We select these candidates because they have been designed to perform well across various important vision tasks.

To select the best combination, we train Theia-T using different sets of teachers and evaluate the learned representation on the MuJoCo subset of tasks. Figure 5 shows the results of different teacher combinations. We start with single VFM teachers and observe that **C** and **V** are the most beneficial. We then distill **All** teacher models altogether and discover that it performs better than four out of five single-teacher distillations, but performs worse than only distilling **V**, suggesting possible negative effects from some of the teacher models. We then remove each teacher model from **All** and distill the remaining four (**All−X**). The results show that removing **V** and **C** from the teachers causes the most significant performance drops while taking out **S** and **De** leads to either similar or improved performance. Given the (nearly) negative effects observed, we distill the rest of the three models (**CDiV**), which results in the best performance. We also try the teacher combination that was performed in RADIO [53] (**CDiS**), but it does not outperform **CDiV**.

**Model Design.** We conducted ablation studies on distillation loss choices and model architecture. We compared two kinds of distillation losses: MSE and Cos+L1 from Ranzinger et al. [53]. As shown in Table 3a, we find that Cos+L1 performs better. For model architecture, we examined the effect of "register tokens" introduced by Darcet et al. [24]. These extra tokens are introduced at the input without supervision and are discarded when using the representation. With the same amount of spatial tokens supervised by distillation, we varied the number of register tokens of between 0, 1, 4, and 8. Note the unsupervised [CLS] token is removed in zero register token case, and it counts for 1 register token in other cases. According to Table 3b, we find that using 1 register token worked the best, while having no register tokens performed the worst. Compared to RADIO [53], which distills both [CLS] and spatial tokens, our use of the [CLS] token as a register token provides an advantage. We also evaluated distilling both [CLS] and spatial tokens in the Theia model, and the results shown in Table 3c confirm the advantage of exclusively distilling spatial tokens.

### 4.5 Qualitative Visualizations

We present qualitative visualizations to demonstrate how Theia-representations can be transformed into teacher representations through feature translators. Using Theia trained with all teachers (**CDeDiSV**, **All**), we applied PCA for visualizing predicted DINOv2 [7] features, used the SAM [10] decoder to produce segmentation results, and used the Depth-Anything [1] head to produce estimated depth. Results are shown in Figure 6 with more examples in the Appendix. The visualizations indicate that our predicted rep-

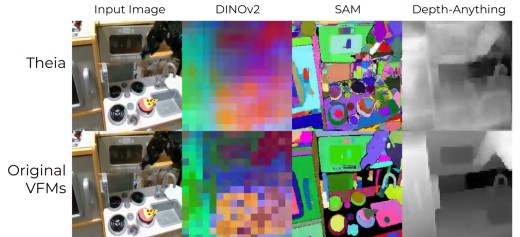

Figure 6: Visualization of VFM outputs from Theia predicted representations (top) and original VFM representations (bottom).

resentations can be decoded by the original VFM and produce reasonable results. Encouragingly, we find that predicted depth maps from the Theia-predicted representation appear to be more accurate compared to the original Depth-Anything model, particularly in the stove-top area and at the microwave door. This shows the potential benefit of combining different visual understandings.

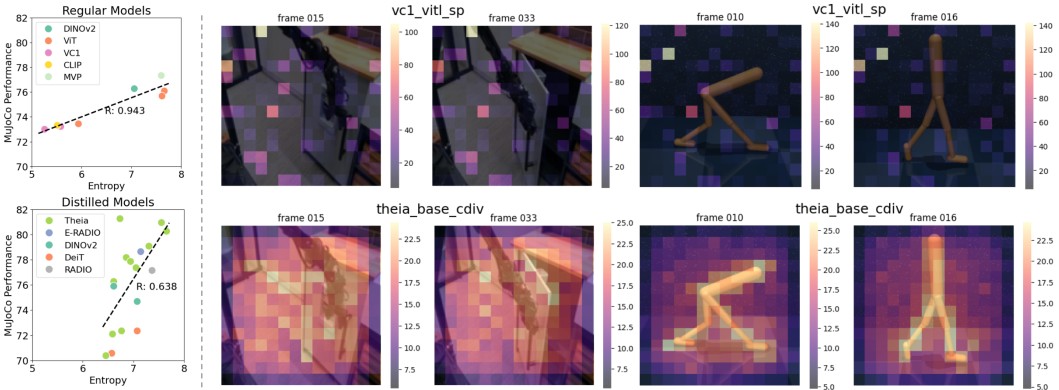

Figure 7: Left: Correlations between feature norm distribution entropy and robot learning performance; Right: Visualizations of spatial token feature norms from VC-1-L-sp and Theia-B.

# 5   What Makes Visual Representations Good for Robot Learning?

Traditionally, the quality of the pre-trained visual representations is evaluated through downstream robot learning like IL or RL. However, it is unclear why different visual representations lead to varying robot learning performance outcomes. In this section, we quantify the quality of visual representations and analyze how they correlate with downstream robot learning performance.

**Feature Norm Distributions and Entropy.**   Darcet et al. [24] analyzed the norm of spatial tokens in vision Transformers and found that high-norm outlier tokens are detrimental to vision task performance. Following this, we investigate whether a similar phenomenon arises in visual representations for robot learning. We inspect the feature norms of Theia with different teacher combinations and baseline models evaluated in Section 4.2, and their corresponding performance on the MuJoCo subset tasks. We sample 1% of the MuJoCo task training set and calculate the L2-norm of each spatial token after encoding images. We measure the entropy of the feature norm distribution across all samples and patches per model and use it as a quantitative metric. To calculate the entropy of the distribution, we first discretize the distribution by a histogram. Then, we use the following formula to obtain the entropy $H = -\sum_i (p_i * \log(p_i))$, where $p_i$ is the probability of each discretized bin.

We confirm that similar outlier tokens also appear in VC-1 corresponding to the image patches that are not task-relevant, shown in the visualizations of feature norms on the right of Figure 7. In contrast, Theia has very few or no outlier tokens, and the tokens with higher norms are more task-relevant even though Theia-representations are not trained on these robot images. In our quantitative analysis (Figure 7, left), we divide the models into distilled and regular based on the observation that distilled models generally have higher entropy (fewer outliers, Figure 4(c) in Darcet et al. [24]). We find that there is a strong correlation (R=0.943) between entropy and robot learning performance among regular models, and a high correlation (R=0.638) among distilled models. We hypothesize that spatial token representations with high entropy (better feature diversity) encode more information that aids policy learning, while less diverse representations (low entropy) may hinder it. In the Appendix G, we discuss the results of other quantitative measurements, including feature similarity and PCA-explained variance ratios, where no strong correlations are found.

# 6   Conclusion

In this work, we introduced Theia, a novel robot vision foundation model specifically distilled from multiple VFMs to enhance robot learning tasks. Theia builds a rich visual representation from diverse VFM teachers, preserving spatial features to ensure detailed visual comprehension. Through extensive evaluations on CortexBench and in the real world, Theia consistently outperforms state-of-the-art models, including all prior models for robot learning, off-the-shelf VFMs, and similarly distilled models for vision tasks. Our results highlight the effectiveness of distilling multiple VFMs into a compact model for superior performance in a variety of robot learning scenarios. Furthermore, we answer a key question about what kinds of visual representations lead to better robot learning by finding a strong correlation between the entropy of feature norms and enhanced downstream performance, offering insights for future research on optimizing visual representations for robotics.

**Acknowledgments**

The authors thank Deva Ramanan for inspiring discussions on the analysis of trained models, Osman Dogan Yirmibesoglu for his help with the Spot grippers, and Joe St. Germain and Jien Cao for their assistance in setting up the WidowX robots. We also extend our gratitude to Erica Lin and Ahmet Gundogdu for their work on Spot's policy learning infrastructure, as well as Nathan Williams and Brennan Vanderlaan for DevOps support.

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

# A Theia Model Architecture

**Backbone.** We use the DeiT-Tiny, DeiT-Small, and DeiT-Base models [42] as our backbone archi-
tectures. We keep the `[CLS]` token in the model and in the forward pass, but there is no supervisory
signal provided for it. As a result, the `[CLS]` token serves as a "register token" [24], which provides
some benefits for learning high quality representations. We train Theia from scratch (no pre-trained
DeiT [42] weights are applied).

**Feature Translators.** The feature translators are composed primarily of CNNs, with a linear
layer appended at the end to match the teacher's representation dimension. Pure linear transforms
might not be able to map Theia-representations to all three teacher representations well, resulting in
a failure of learning (See Table 6a). Thus, we use three CNN layers to account for the fact that each
teacher model's representations are very different from one another. Details are listed in Table 4,
where we show the architectural details of the translators used for our (student, teacher)-feature
pairs.

Table 4: Feature Translator configurations

| Student $d_s \times 14 \times 14$ (Theia backbone=ViT-T, -S, -B) to Teacher $d_t \times 16 \times 16$ (CLIP, DINOv2, ViT) |
|---|
| ConvTranspose2d($d_s$, $d_s$, kernel_size=3, stride=1, output_padding=0) |
| LayerNorm |
| Conv2d($d_s$, $d_s$, kernel_size=3, padding=1) |
| ReLU+LayerNorm |
| Conv2d($d_s$, $d_s$, kernel_size=3, padding=1) |
| ReLU+LayerNorm |
| Flatten and Linear($d_s$, $d_t$) |

| Student $d_s \times 14 \times 14$ to Teacher $d_t \times 64 \times 64$ (SAM and Depth-Anything) |
|---|
| ConvTranspose2d($d_s$, $d_s$, kernel_size=3, stride=2, padding=1 |
| LayerNorm |
| ConvTranspose2d($d_s$, $d_s$, kernel_size=3, stride=2, output_padding=1) |
| ReLU+LayerNorm |
| Conv2d($d_s$, $d_s$, kernel_size=3, padding=1) |
| ReLU+LayerNorm |
| Flatten and Linear($d_s$, $d_t$) |

**Feature Normalization.** Formally, the normalization is:

$$h(\tilde{\mathbf{x}}_j)_c = \frac{h(\mathbf{x}_j)_c - \mu_c}{\sigma_c}, \ \mu_c = \frac{1}{N} \sum_j h(\mathbf{x}_j)_c, \ \sigma_c = \sqrt{\frac{\sum_j (h(\mathbf{x}_j)_c - \mu)^2}{N}}, \tag{2}$$

where $c$ is the channel index for the feature, $j$ is the index of image sample, and $N$ is the number of
samples in ImageNet.

# B Training

We train Theia on 8 NVIDIA H100 GPUs. The main bottlenecks in training are the data transfer
speed between devices and the GPU memory bandwidth to load large spatial feature tensors, for
example, of size $1280 \times 16 \times 16$ for ViT-H and $256 \times 64 \times 64$ for SAM. We pre-compute the features
from all teacher models instead of doing inference on the fly. This approach requires extra storage
space to save all the features extracted from the VFMs, but significantly saves on training time and
avoids loading models with high GPU memory usage during training, such as Depth-Anything or
SAM (a batch size of 16 cannot fit into 80GB of GPU memory). All training configurations are
listed in Table 5.

**Teacher VFM Features.** We use the output representations at the last layer of ViT [5], CLIP [2],
and DINOv2 [7]. For SAM [10], we use its encoder output. For Depth-Anything [1], since it is

Table 5: Theia Training Configuration

| Hyperparameters | |
| --- | --- |
| # GPUs | 8 |
| Batch size | 16 / GPU (128 effective) |
| Learning rate (LR) | 5e-4 |
| LR Schedule | Constant |
| Weight decay | 0.01 |
| Optimizer | AdamW |
| Betas | [0.9, 0.999] |
| Epochs | 50 |
| Warm-up epochs | 5 |
| Warm-up LR schedule | Linear (1e-2*LR) |
| Gradient clipping | None |
| Image augmentation | None |
| Total GPU hours | 152 |

Table 6: More ablation studies on model design. All experiments are based on Theia-T. Scores are the average performance from the MuJoCo subset of CortexBench.

(a) Feature Translator Architecture

| CNNs | Linear |
| --- | --- |
| **78.9** $\pm$ 0.8 | 41.7 $\pm$ 1.6 |

(b) Training from Scratch vs Pre-trained Backbone

| Training from Scratch | Pre-trained Backbone |
| --- | --- |
| 78.9 $\pm$ 0.8 | **80.8** $\pm$ 1.5 |

initialized from DINOv2, we use the latent representation before the final convolution layer. When decoding SAM and Depth-Anything results from Theia-predicted representations, we send the predicted representations through the remaining layers of original models and obtain the output.

## C  Additional Ablation Studies

We conduct two additional ablation studies to verify design choices in the Theia model. The first is a comparison between the current CNN-based feature translator and a linear feature translator. In Table 6a, we find that using a Linear feature translator leads to a significant performance drop. The second ablation studies whether Theia should be trained from scratch or be initialized using the pre-trained DeiT [42] weights. In Table 6b, we find that using pre-trained weights improves the downstream performance. This could be interpreted as the positive effect of incorporating knowledge from an additional useful model into the distillation process. We would expect to see similar performance improvements as more informative models are included during training.

## D  Full Experimental Settings

Table 7: Comparison of model architectures, training datasets, total number of images, objectives, and training duration (epochs or GPU hours) across the models used in this paper. We use the numbers reported in their original papers and - stands for we could not find such information.

| Model | Architecture | Dataset(s) | Total # Images | Objective | Training Duration |
| --- | --- | --- | --- | --- | --- |
| Theia | ViT | ImageNet-1k [23] | 1.2M | Distillation | 50 epochs / 152 GPU hours on H100s |
| RADIO / E-RADIO [53] | ViT/Self-designed | DataComp-1B [73] | 1.4B | Distillation | - |
| VC-1 [20] | ViT/MAE [14] | ImageNet-1k [23]+V [33, 54, 56, 55]+N | 5.6M | MAE [14] | 182 epochs / over 10,000 GPU hours |
| MVP [19] | ViT/MAE [14] | ImageNet-1k [23]+Video [56, 55, 54] | 1.9M | MAE [14] | 1600 epochs |
| R3M [17] | ResNet [4] | Ego4D [33] | - | Time Contrastive [35]+Vision-Language Alignment | 1.5M steps |
| VIP [18] | ResNet [4] | Ego4D [33] | 4.3M | VIP [18] | - |
| DINOv2 [7] | ViT | LVD-142M [7] | 142M | Self-distillation | 22,016 GPU hours for DINOv2-g |
| CLIP [2] (Vision Encoder) | ViT | - | 400M | image-text contrastive learning [2] | 73,728 GPU hours for CLIP ViT-L/14 on V100s |
| ViT [5] | ViT | ImageNet-21k [23] / JFT-300M | 14M / 300M | Classification | 90 epochs / 7 epochs |
| DeiT [42] | ViT | ImageNet-1k | 1.2M | Classification+Distillation | 300 epochs / 288 GPU hours on V100s |

### D.1 Baseline Models

Theia and baseline models are trained on different sizes of datasets using different objectives. We organize these details in Table 7 to provide a comprehensive comparison between them.

### D.2 CortexBench

For all of our CortexBench experiments, we use the original setup [20], except for a few modifications to produce more reliable results. The modifications include:

- We increase the number of evaluation roll-outs from 10 (original) to 25 (ours) in DMC tasks. The mean scores reported are from a total of 75 runs (25 per seed x 3).

- We remove the noise added to the policy network output in the CortexBench code base. The noise causes minor performance degradation (about 1.0 on overall mean score for MuJoCo tasks) compared to the version without noise.

- We modify the policy networks to take spatial feature inputs for MuJoCo and Trifinger tasks (details follow and are presented in Table 8).

Note that prior models including R3M, VIP, MVP, and VC-1 are all re-run using the same settings in MuJoCo tasks for the purposes of making a fair comparison when evaluating against Theia.

**Policy Networks.** For MuJoCo and Trifinger tasks we utilize a three-layer MLP for vector-based representations, including ResNet models and Transformer models that use the `[CLS]` token. For models that generate spatial feature maps, such as Transformers using spatial tokens, we introduce a three-layer CNN before the MLP. For Habitat tasks, we exclusively benchmark models that produce spatial feature maps and adopt the same policy network as used by Majumdar et al. [20]. Details can be found in Table 8.

Table 8: Policy Networks for MuJoCo Tasks

| Spatial Representation dimension $d \times H \times W$ |
|---|
| Conv2d($d$, 256, kernel_size=4, stride=2, padding=1) |
| ReLU |
| Conv2d(256, 256, kernel_size=3, stride=2) |
| ReLU |
| Conv2d(256, 256, kernel_size=3, stride=1) |
| Flatten and Linear(256, 256) |
| Linear(256, 256) |
| Linear(256, action dimension) |
| Vector Representation dimension $d$ |
| Linear($d$, 256) |
| Linear(256, 256) |
| Linear(256, action dimension) |

For ImageNav task, we use the provided policy network from VC-1 [20] without modification. The policy network is composed by a compression-layer (a CNN layer) to convert the spatial feature map into a vector representation, followed by a 2-layer LSTM. Details are available in Appendix A.2 in [20].

### D.3 Real World Robot Learning

### D.3.1 WidowX Arm Experiments

**WidowX Arm Setup.** The robot used for these experiments is a 6-degree-of-freedom (DOF) WidowX 250s arm. The data collection and evaluation framework is based on [59].

Table 9: Number of demonstrations and data collection method for the real-world experiments.

| Task | Door-Opening | Pick-and-Place | Toy-Microwave Cooking | Drawer Opening |
|---|---|---|---|---|
| # Demos | 48 | 63 | 101 | 50 |
| Method | Teleoperation | Teleoperation | Teleoperation | Scripted Policy |

Table 10: WidowX Policy Training Configuration

| Hyperparameters | |
|---|---|
| Batch size | 16 |
| Learning rate | 1e-4 |
| Weight decay | 0.01 |
| Optimizer | AdamW |
| Betas | [0.9, 0.999] |
| Epochs | 400 |
| Loss | SmoothL1 |

We train a behavior-cloning policy for each of the four evaluated setups and for each evaluated baseline (see Table 2 and Figure 3); the training hyperparameters are shown in Table 10. To train each of the three tasks performed with the WidowX robot, i.e., Door Opening, Pick-and-Place, and Toy-Microwave Cooking, we collected human demonstrations by teleoperating the robot with a Virtual Reality (VR) controller using the setup introduced in [59]; the number of collected demonstrations for each of the tasks is reported in Table 9. The policy's observations are RGB images and robot joint states. Images are encoded by a pre-trained visual encoder and a randomly initialized, unfrozen feature neck ("compression layer" [71]). We use the same feature neck as we did for the previously discussed MuJoCo tasks in Section 4.2. The encoded vector is concatenated with the robot's joint states, which is fed into a 3-layer MLP with a hidden dimension of size 256. The policy outputs end-effector commands, consisting of the end-effector's delta positions (Cartesian coordinates), delta rotations (Euler angles), and the gripper's opening/closing command; such commands are tracked by the robot at a frequency of 5 Hz. In addition to the hyperparameters listed in Table 10, we vary the policy action prediction horizon depending of the difficulty of the task, i.e., at each step the policy predicts the next 10, 10 and 5 actions for Door Opening, Toy-Microwave Cooking, and Pick-and-Place, respectively.

In the following, we give more details about the WidowX tasks showcased in our work.

**Door Opening.** In this task the robot has to open a fridge door in a toy-kitchen setup; we identify two stages to evaluate the task's success: *Open* and *Fully Open* (see Figure 3). We place the robot in front of the fridge and collect 63 demonstrations to train the behavioral cloning policy. We vary the height (z-axis) of the robot base between 40-46cm, and the position (x-axis, parallel to the toy kitchen) of the robot base; samples from the initial state distribution of the demonstrations are shown in Figure 8. At inference time, we vary the height of the robot base among $\{40, 42, 43, 44, 46\}$ cm, and select between 5 randomly-picked positions along the x-axis (for all policies). For each robot position, we evaluate the policy twice, for a total of 50 runs.

**Pick-and-Place.** In this task, the robot has to pick up a pink cup from a toy-sink and drop it into a drying rack located on the left of the sink. We collected 48 demonstrations to train the policy, where we varied the initial pose of the objects, as shown in Figure 8. During evaluation, we vary the cup's starting position amongst a total of 10 positions, of which 8 positions are equally distributed about the perimeter, and 2 are in the center of the sink. We also roughly vary the direction of the cup handle towards the left or the right. In total, we evaluate this task for 20 runs. There are two key stages for which we measure the success rate: picking up the cup and successfully releasing it into the drying rack.

**Toy-Microwave Cooking.** In this task, the robot has to pick up an object from within the pot on the stovetop, putting the object into a toy-microwave, and closing the microwave. In each test, we

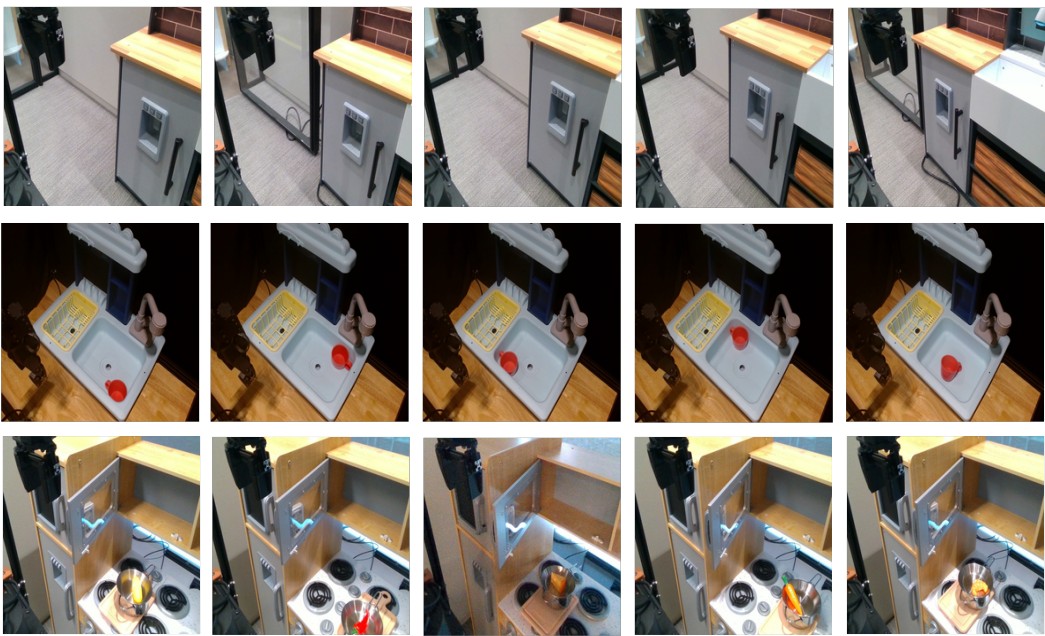

Figure 8: Samples from the initial state distribution of the collected demonstrations for *fridge door-opening* (top), *pick-and-place* (middle), and *toy-microwave cooking* (bottom). For each of these tasks, the factors of variation are, respectively: the robot height and position with respect to the door, the initial pose of the cup, and the initial position of the objects and object types.

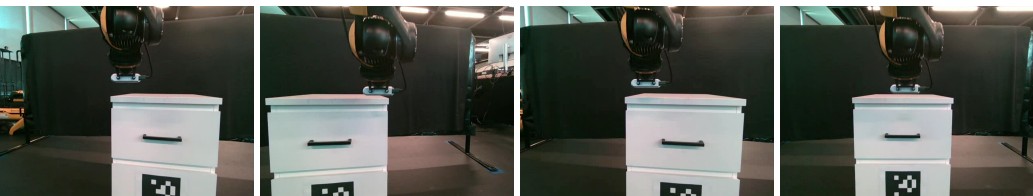

Figure 9: Samples from the initial state distribution of the collected demonstrations for the Spot drawer opening experiments. For this task, the factors of variation are the position and orientation of the robot with respect to the drawer.

initialize the environment with the microwave door open. In this task, we collected 100 demonstrations across 10 different toy-food objects (10 demonstrations per object) with randomized object positions; examples from the initial state distributions of the collected demonstrations can be seen in Figure 8. During evaluation, we test 40 runs on 10 seen, in-distribution objects (4 runs per object), and 10 runs on 5 unseen, out-of-distribution objects (2 each), for a total of 50 runs. Furthermore, we vary the position of the pot that holds the object. We point out that this task is characterized by a longer horizon with respect to the other two; in fact, it has three different steps: 1. picking up the object, 2. placing the object into the microwave, 3. and closing the microwave. In addition, the policy needs to generalize to the different poses of the objects on the stove-top and to out-of-distribution objects. The longer horizon and the variety of objects in this task make it particularly challenging, so using frozen visual encoders was not effective (0% success rate). However, with fine-tuning, the policies performed reasonably well.

### D.3.2 Spot Experiments

**Spot Setup.** For the Spot experiments, we train a diffusion policy [72] conditioned on the encoded image. The diffusion policy outputs the desired absolute positions and rotations of the end-effector and the gripper state. Hyperparameters for the policies are shown in Table 11.

Table 11: Spot Policy Training Configuration

| Hyperparameters | |
|---|---|
| Batch size | 32 |
| Learning rate | 3e-4 |
| Weight decay | cosine |
| Optimizer | Adam |
| Betas | [0.9, 0.999] |
| Training Iterations | 3000 |
| Loss | MSE |
| Policy Horizon | 4 |

**Drawer Opening.** In this task, the robot has to open the top drawer of a cabinet. The policy receives color images from a single forward facing camera mounted on the body of the Spot. A fiducial marker is used to enable the robot to walk to a random position and orientation for each trial. Samples from the distribution of random initial states are shown in Figure 9. The starting locations vary by $\pm5$ cm in x and y and $\pm0.2$ radians in orientation. 50 successful demonstrations were collected using a scripted policy and we evaluate each policy for 20 trials. A trial is considered successful if the drawer is opened at least 10 cm. After each trial, a scripted policy uses the fiducial marker to reset the environment and the robot moves to a new random location.

# E  More Visualizations of Translating to Teacher Model Features

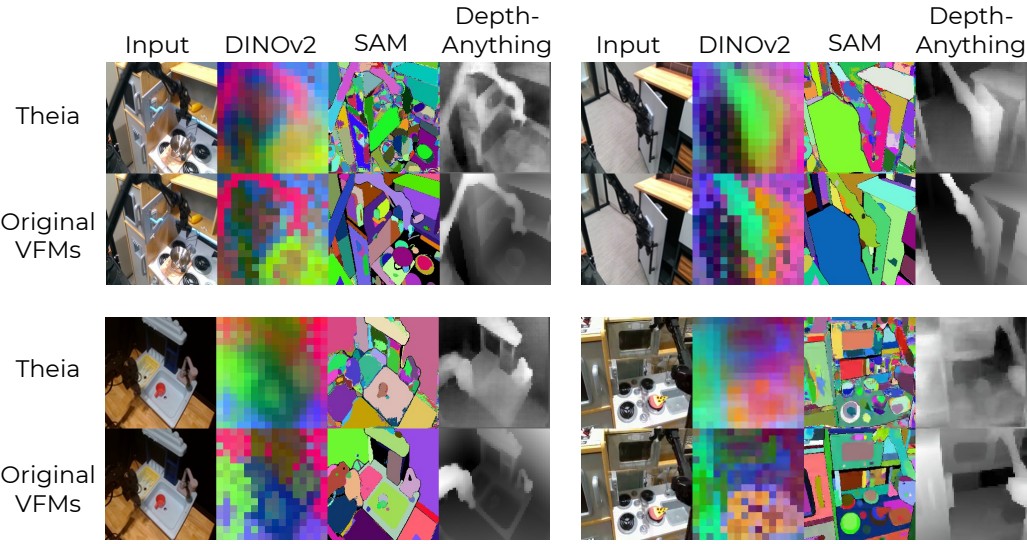

Figure 10: More examples of decoding Theia-representation to VFM outputs using feature translator and original VFM decoders. We select robot images from our experiment recordings. Theia and VFMs are not trained on these images.

We attach examples of decoding Theia-representations of 4 frames from a robot video into VFM outputs in Figure 10. Note that Theia and VFMs are not trained on the robot images on which we run this evaluation.

# F  Per-Task CortexBench Results

In Table 12 we report per-task scores of the models evaluated in Figure 4, over the MuJoCo subset of tasks. In Table 13, we report per-task scores of all 14 tasks we evaluated in Cortexbench, corresponding to Table 1. Note that we perform the evaluation following the original Cortexbench [20]

protocol, where there are a total of 75 runs per MuJoCo task (we increased it from 30 to 75 for DMC tasks), 100 runs in Reach Cube, and 4200 runs in ImageNav.

Table 12: Per-Task Results on the MuJoCo Subset.

| Model | Assembly | Bin-picking | Button-press | Cheetah-run | Finger-spin | Reacher-easy | Walker-stand | Walker-walk | Drawer-open | Hammer | Pen | Relocate |
|---|---|---|---|---|---|---|---|---|---|---|---|---|
| Theia-T | 90.00±8.49 | 74.00±8.49 | 80.00±0.00 | 63.78±1.65 | 69.76±0.97 | 79.18±3.87 | 90.59±1.60 | 81.06±1.92 | 100.00±0.00 | 98.00±2.83 | 74.00±2.83 | 46.00±2.83 |
| Theia-S | 94.67±9.24 | 70.67±11.55 | 72.00±14.42 | 67.37±4.25 | 70.46±0.80 | 83.25±5.19 | 92.24±0.75 | 82.62±1.91 | 100.00±0.00 | 97.33±4.62 | 81.33±2.31 | 50.67±6.11 |
| Theia-B | 93.33±8.33 | 76.00±4.00 | 82.67±6.11 | 67.67±1.92 | 70.84±1.37 | 83.23±7.05 | 92.55±3.67 | 81.33±3.11 | 100.00±0.00 | 98.67±2.31 | 78.67±2.31 | 46.67±2.31 |
| DINOv2-L | 93.33±8.33 | 80.00±8.00 | 61.33±2.31 | 45.66±5.69 | 70.95±0.25 | 74.24±16.02 | 92.84±4.55 | 83.70±1.34 | 100.00±0.00 | 100.00±0.00 | 77.33±4.62 | 36.00±0.00 |
| DINOv2-B | 92.00±8.00 | 76.00±14.42 | 72.00±4.00 | 48.02±3.71 | 70.77±0.59 | 75.84±2.63 | 92.64±1.81 | 83.82±2.26 | 100.00±0.00 | 98.67±2.31 | 68.00±4.00 | 33.33±2.31 |
| DINOv2-S | 93.33±8.33 | 68.00±8.00 | 81.33±12.22 | 45.43±5.33 | 70.70±0.81 | 59.86±7.73 | 88.21±1.90 | 77.62±6.32 | 100.00±0.00 | 98.67±2.31 | 77.33±4.62 | 36.00±4.00 |
| CLIP-L | 69.33±4.62 | 76.00±4.00 | 64.00±8.00 | 33.63±1.21 | 69.97±2.02 | 89.42±3.92 | 95.12±0.89 | 75.87±5.33 | 100.00±0.00 | 96.00±4.00 | 73.33±8.33 | 37.33±6.11 |
| ViT-H | 94.67±9.24 | 76.00±12.00 | 77.33±9.24 | 47.89±10.10 | 69.84±1.16 | 84.33±6.46 | 90.97±5.36 | 79.99±6.24 | 100.00±0.00 | 94.67±2.31 | 56.00±0.00 | 41.33±4.62 |
| ViT-L | 96.00±5.66 | 54.00±25.46 | 81.60±3.58 | 50.32±5.24 | 69.54±0.80 | 84.49±3.26 | 89.43±2.30 | 77.43±1.80 | 100.00±0.00 | 96.00±3.27 | 68.00±3.27 | 41.60±8.29 |
| ViT-B | 96.00±6.93 | 74.67±10.07 | 76.00±10.58 | 46.28±5.32 | 72.05±1.23 | 73.71±0.80 | 77.13±6.50 | 69.75±3.36 | 100.00±0.00 | 96.00±4.00 | 68.00±0.00 | 32.00±4.00 |
| ViT-S | 93.33±8.33 | 77.33±2.31 | 61.33±6.11 | 42.99±1.13 | 70.71±0.68 | 70.46±2.59 | 88.34±2.63 | 67.83±4.00 | 100.00±0.00 | 90.67±9.24 | 68.00±4.00 | 37.33±4.62 |
| ViT-T | 93.33±8.33 | 82.67±4.62 | 65.33±10.07 | 39.02±1.47 | 71.45±1.07 | 71.13±3.24 | 84.05±0.84 | 70.74±3.03 | 100.00±0.00 | 84.00±6.93 | 60.00±10.58 | 25.33±4.62 |
| VC-1-L-sp | 85.33±8.33 | 66.67±12.22 | 56.00±8.00 | 66.88±6.66 | 71.19±0.67 | 70.67±8.36 | 93.43±6.08 | 83.28±2.40 | 100.00±0.00 | 93.33±2.31 | 68.00±0.00 | 24.00±8.00 |
| CDV | 73.33±24.44 | 74.40±11.87 | 30.67±39.26 | 50.53±12.08 | 71.94±1.07 | 71.60±17.03 | 94.85±1.27 | 78.36±5.72 | 100.00±0.00 | 93.60±6.07 | 75.33±5.89 | 39.33±10.25 |
| RADIO | 96.00±6.93 | 84.00±8.00 | 82.67±12.86 | 35.92±1.59 | 71.98±1.41 | 78.46±6.50 | 89.06±2.42 | 81.33±4.91 | 100.00±0.00 | 98.67±2.31 | 68.00±0.00 | 40.00±6.93 |
| E-RADIO | 94.67±9.24 | 82.67±2.31 | 80.00±4.00 | 57.37±2.27 | 69.68±1.05 | 75.59±1.29 | 91.73±2.03 | 80.32±2.06 | 100.00±0.00 | 100.00±0.00 | 66.67±12.86 | 45.33±2.31 |
| MVP-L-sp | 93.33±8.33 | 73.33±4.62 | 82.67±10.07 | 68.07±1.71 | 71.03±2.10 | 69.87±6.75 | 88.44±4.21 | 80.14±1.50 | 100.00±0.00 | 97.33±2.31 | 77.33±12.22 | 26.67±11.55 |
| MVP-L | 94.67±9.24 | 82.67±9.24 | 89.33±8.33 | 34.62±5.83 | 68.63±2.02 | 67.95±3.18 | 74.50±1.65 | 48.04±1.37 | 100.00±0.00 | 88.00±6.93 | 62.67±6.11 | 20.00±4.00 |
| R3M | 96.00±6.93 | 92.00±4.00 | 68.00±4.00 | 55.88±1.12 | 70.65±0.34 | 82.37±3.70 | 88.88±2.70 | 69.52±4.94 | 100.00±0.00 | 98.67±2.31 | 73.33±2.31 | 58.67±4.62 |
| VIP | 93.33±8.33 | 70.67±8.33 | 76.00±4.00 | 45.10±4.02 | 69.02±0.67 | 68.08±3.45 | 78.50±2.49 | 63.52±1.40 | 98.67±2.31 | 96.00±4.00 | 73.33±6.11 | 29.33±10.07 |

# G   Analysis of Visual Representations

**Entropy of the Representation Norm Distribution.**   Given $N$ representations produced by encoding $N$ images per model, where each representation contains $P$ spatial tokens, we discretize the distribution of token norms over all $N \times P$ tokens by using a histogram. We normalize the count of each bin in histogram by the total number of tokens to obtain the probabilities of each bin. We then calculate the Shannon entropy, given by $-\sum_i p_i \log(p_i)$. We find that the distilled models have higher entropy than the regular models, so we divide them into two distinct groups. Results are plotted as model performance vs entropy on the MuJoCo tasks. We attach the full version of plot presented on the left of Figure 7 here in Figure 11, including two plots corresponding to each category of models and one plot for all models.

At the top of Figure 12, we find that both CLIP and VC-1 have high-norm outlier tokens. To better visualize the values of normal tokens, we use the median of norm values to clip the values. Specifically, we clip the norm values to range $[0, 2 * \text{median}]$ and visualize the clipped norm values on the bottom of Figure 12. We find that the high-norm tokens are still not task-relevant.

In Figure 13, we attach the feature norm map of all other models. Among those, we find that MVP [19], which performs well on CortexBench, also produces features without outlier tokens. Feature norms from Depth-Anything [1] and SAM [10], in contrast, have low diversity.

We investigated the distributions of the feature norms that could give more information. Plots in Figure 14 we show the histogram of the feature norm entropy of 4 selected models: ViT-B, DINOv2-B, VC-1-B, and Theia-B (refered to as rvfm_deit_base_4features in the legend).

We find that Theia has the most balanced distribution of the feature norm, while the others are long-tailed. ViT-B has two peaks in the histogram. VC-1 is extremely centered and heavily long-tailed. DINOv2 is relatively less long-tailed than ViT-B and VC-1, and has a relatively nicer distribution around the median value.

**PCA Explained-Variance Ratio of Representations.**   Similar to the entropy analysis, given $N \times P$ spatial token representations, we apply PCA to them and extract the explained-variance ratio (EVR) of each latent dimension. We calculate and plot the cumulative sum of EVRs, as well as calculate the Area Under the Curve (AUC) of cumulative sum of the EVR. When comparing Theia-B with ViT-B, DINOv2-B and VC-1-B (Figure 15), we find that Theia-B has the lowest AUC and the best MuJoCo performance, while VC-1 has the highest AUC and the worse MuJoCo performance amongst these 4 models. The higher AUC is caused by one or few principle components that have very high EVRs, indicating that these components are capturing the majority of the variance of the feature representations. This means that less information is encoded within such representations. In contrast, the Theia-representation has a low AUC which we believe is due to the rich information that has been encoded within the latent space.

Table 13: Per-task results on CortexBench (excluding Move Cube, ObjectNav, and MobilePick due to reproducibility issues).

| Model | Assembly | Bin-picking | Button-press | Cheetah-run | Finger-spin | Reacher-easy | Walker-stand | Walker-walk | Drawer-open | Hammer | Pen | Relocate | Reach-cube | ImageNav |
|-------|----------|-------------|--------------|-------------|-------------|--------------|--------------|-------------|-------------|--------|-----|----------|------------|----------|
| Theia-B | 93.33±8.33 | 76.00±4.00 | 82.67±6.11 | 67.67±1.92 | 70.84±1.37 | 83.23±7.05 | 92.55±3.67 | 81.33±3.11 | 100.00±0.00 | 98.67±2.31 | 78.67±2.31 | 46.67±2.31 | 86.19±0.11 | 59.3±0.7 |
| VC-1-L-sp | 85.33±8.33 | 66.67±12.22 | 56.00±8.00 | 66.88±6.66 | 71.19±0.67 | 70.67±8.36 | 93.43±6.08 | 83.28±2.40 | 100.00±0.00 | 93.33±2.31 | 68.00±0.00 | 24.00±8.00 | 84.79±0.63 | 70.3±0.7 |
| E-RADIO | 94.67±9.24 | 82.67±2.31 | 80.00±4.00 | 57.37±2.27 | 69.68±1.05 | 75.59±1.29 | 91.73±2.03 | 80.32±2.06 | 100.00±0.00 | 100.00±0.00 | 66.67±12.86 | 45.33±2.31 | 87.81±0.12 | 53.0±0.7 |
| MVP-L-sp | 93.33±8.33 | 73.33±4.62 | 82.67±10.07 | 68.07±1.71 | 71.03±2.10 | 69.87±6.75 | 88.44±4.21 | 80.14±1.50 | 100.00±0.00 | 97.33±2.31 | 77.33±12.22 | 26.67±11.55 | 87.54±0.2 | 68.1±0.7 |
| R3M | 96.00±6.93 | 92.00±4.00 | 68.00±4.00 | 55.88±1.12 | 70.65±0.34 | 82.37±3.70 | 88.88±2.70 | 69.52±4.94 | 100.00±0.00 | 98.67±2.31 | 73.33±2.31 | 58.67±4.62 | 86.5 | 30.6±0.7 |
| VIP | 93.33±8.33 | 70.67±8.33 | 76.00±4.00 | 45.10±4.02 | 69.02±0.67 | 68.08±3.45 | 78.50±2.49 | 63.52±1.40 | 98.67±2.31 | 96.00±4.00 | 73.33±6.11 | 29.33±10.07 | 86.2 | 48.8±0.8 |

However, when extending the scope to encompass all the models we evaluated (Figure 16 left), we find that the AUC of the EVR does not have a strong correlation with robot learning performance.

**Cosine Similarity of Representations.** We also use cosine similarity to analyze the representations from different models by first calculating the mean of all representations and then computing cosine similarity between each representation and this mean representation. Results are shown on the right of Figure 16, which shows very weak correlation between cosine similarity and performance on CortexBench.

# H Linear Probing on ImageNet

In addition to robot learning, we evaluate the Theia-representation on vision tasks to show to how well such abilities are maintained after the distillation process. For example, to evaluate image classification performance we apply linear probing on the Theia-representation to classify images from ImageNet-1k [23]. We use mean pooling of the Theia-representation (i.e. spatial tokens) and the same training schedule as MAE [14]. Results are shown in Table 14, where we find that Theia outperforms MAE [14] at the same model size, but is not comparable to SOTA results from models like DINOv2 [7].

Table 14: ImageNet-1k [23] evaluation accuracy using linear probing.

| Model | Accuracy |
|-------|----------|
| Theia-B | 72.1% |
| Theia-B (initialized from DeiT-B [42] weights) | 75.2% |
| MAE (ViT-B) [14] | 67.5% |
| DINOv2 [7] (ViT-L) | 84.5% |

# I Probing 3D-awareness of Theia-representation

We evaluate Theia-representation on 3D awareness to show the learned diverse vision knowledge in the representation. In particular, we adopt probe3D [74], an evaluation toolbox to investigate the 3D-awareness of pretrained models. This evaluation includes Monocular Depth Estimation, Surface Normal Estimation, and Multiview Correspondence. We present the evaluation results in Tables 15, 16 and 17, and we summarize them here:

- Theia achieves the third best Depth Estimation performance among over 10 types of vision models (only worse than DeiT-III and DINOv2 models) (Table 2 in Appendix of probe3D [74], NYU dataset).

- Theia achieves comparable performance as the best performing model, DINOv2, on Multiview Correspondence on ScanNet (Table 4, Block3 in Appendix of probe3D [74]).

- Theia has better Surface Normal Estimation performance than CLIP (Table 3 in Appendix of probe3D [74], NYU dataset).

We note that the Theia model is trained only on ImageNet with 224x224 resolution images, while many baseline models in this benchmark are trained on billions of images and higher resolutions. Theia achieves better performance than models with similar training data and resolution.

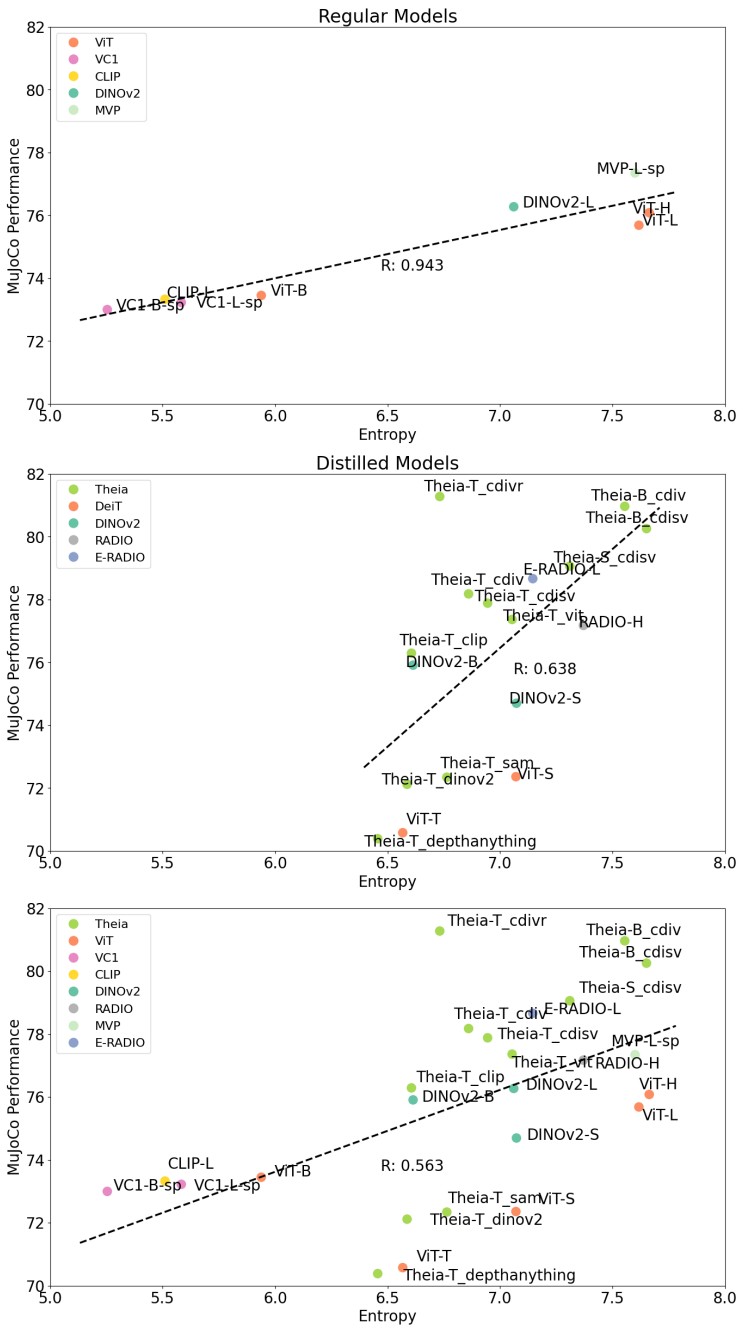

Figure 11: Full results of feature norm entropy.

Table 15: Depth Estimation on NYU dataset

| Model | $\delta_1$ | $\delta_2$ | $\delta_3$ | RMSE |
|-------|-----------|-----------|-----------|--------|
| Theia | 0.8420 | 0.9699 | 0.9929 | 0.4465 |

Table 16: ScanNet Multi-view Correspondence

| Model | $\theta_0^{15}$ | $\theta_{15}^{30}$ | $\theta_{30}^{60}$ | $\theta_{60}^{180}$ |
|-------|-----------------|---------------------|---------------------|----------------------|
| Theia | 47.88 | 37.47 | 24.78 | 13.18 |

Table 17: Surface Normal Estimation on NYU dataset

| Model | 11.25° | 22.5° | 30° | RMSE |
|-------|--------|-------|-----|--------|
| Theia | 30.22 | 54.22 | 65.13 | 33.59 |

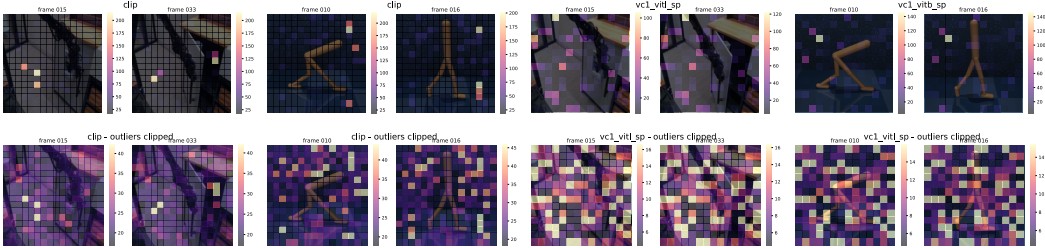

Figure 12: Feature norm map visualizations of CLIP and VC1, original (top) and with high-norm outlier values clipped (bottom)

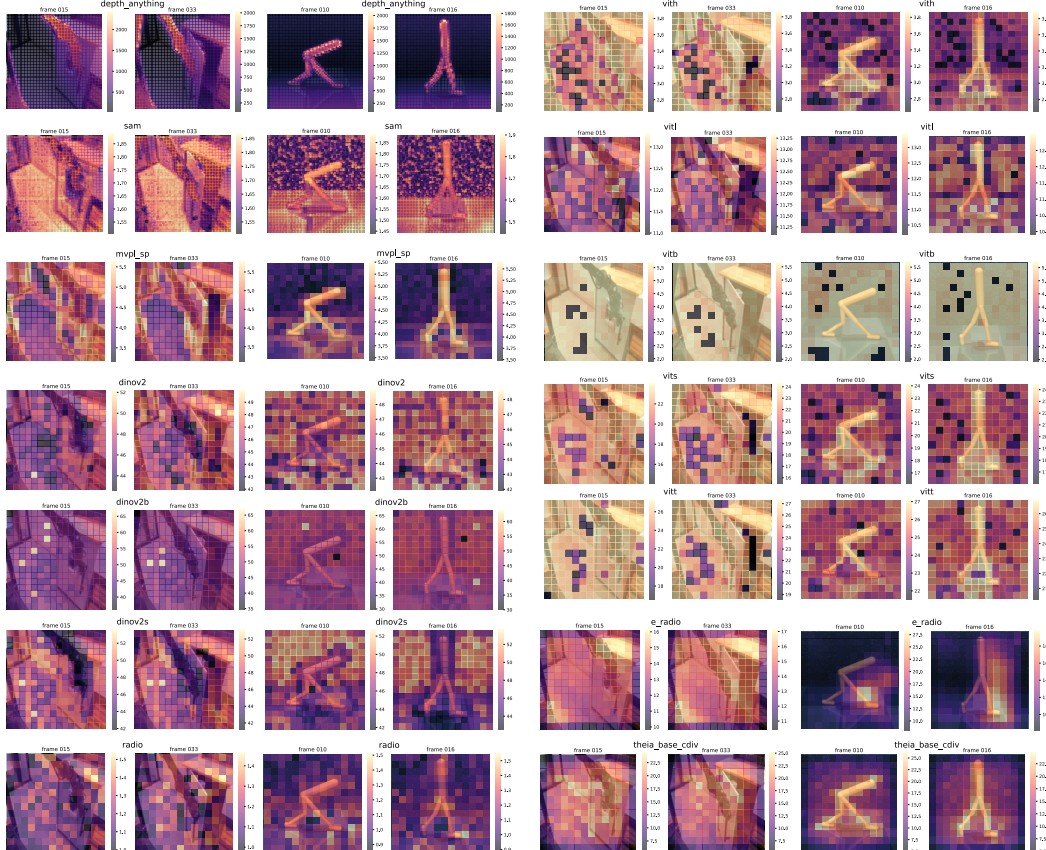

Figure 13: Feature norm map visualizations of ViT [5], DINOv2 [7], MVP [19], Depth-Anything [1], SAM [10], RADIO [53], and E-RADIO [53].

## J Generalization Evaluation

We evaluate Theia-B and baseline models on a simulated benchmark, Factor-World [75], to test the generalization ability of the pre-trained visual representations. We use the Door-Open task from Factor-World benchmark environment, for which we collected 100 trajectories across 20 randomized environments. We trained the policies using the same policy head that we use in CortexBench MuJoCo tasks (Table 8), and we evaluated the generalization ability when varying each of 6 factors of variations over 20 different runs controlled by one factor at a time. In Table 18, we report the success rates of the baselines when varying each factor. These results show that Theia-B has the best generalization abilities compared to R3M, E-RADIO, and MVP.

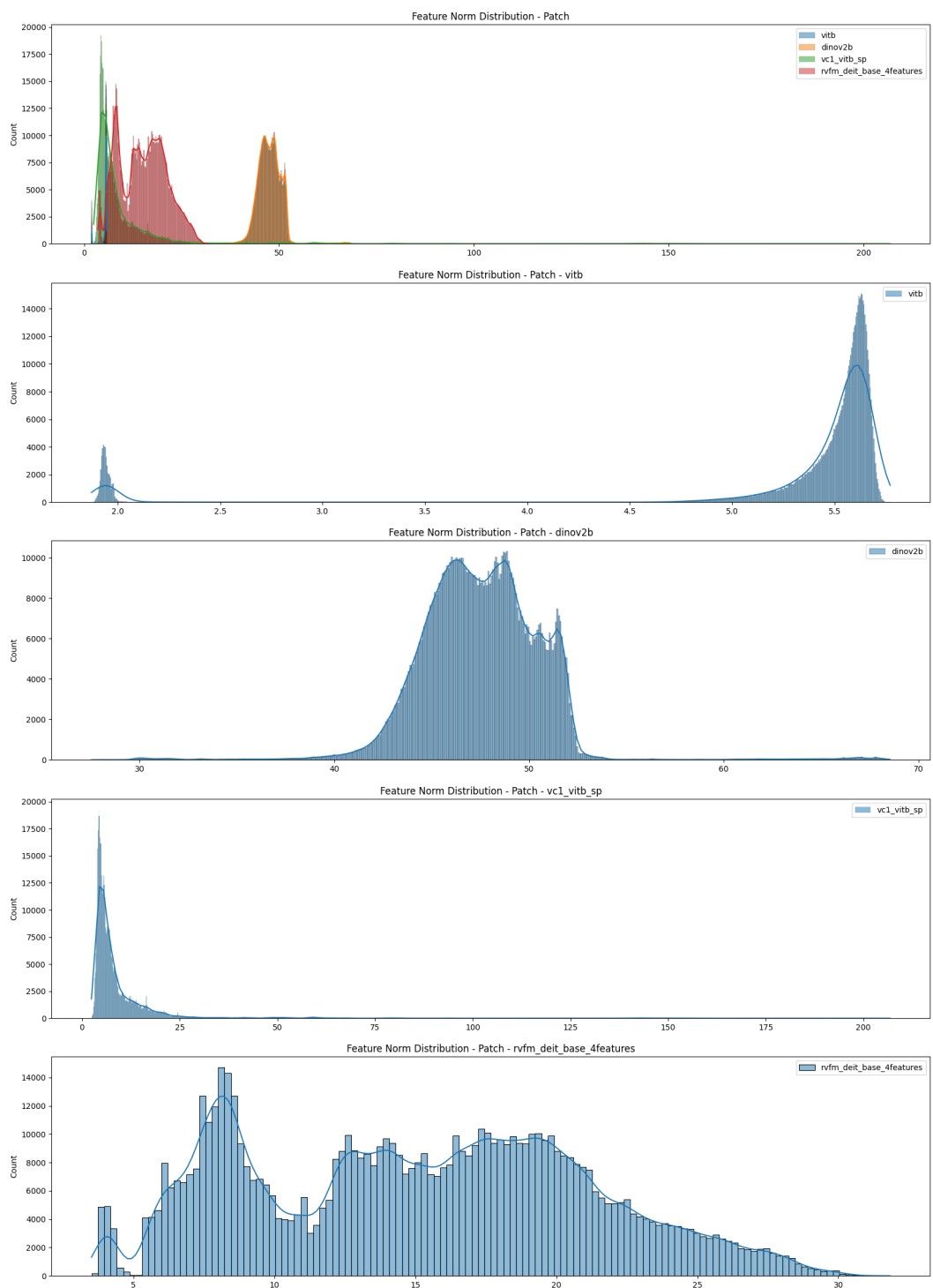

Figure 14: Distrubtions of feature norms. From top to bottom: 4 models on the same plot, ViT-B, DINOv2-B, VC-1-B, and Theia-B.

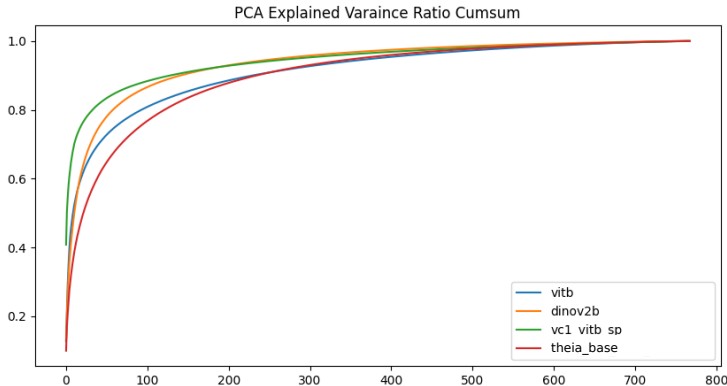

Figure 15: Cumulative sum of PCA Explained Variance Ratio of features from ViT-B, DINOv2-B, VC-1-B, and Theia-B.

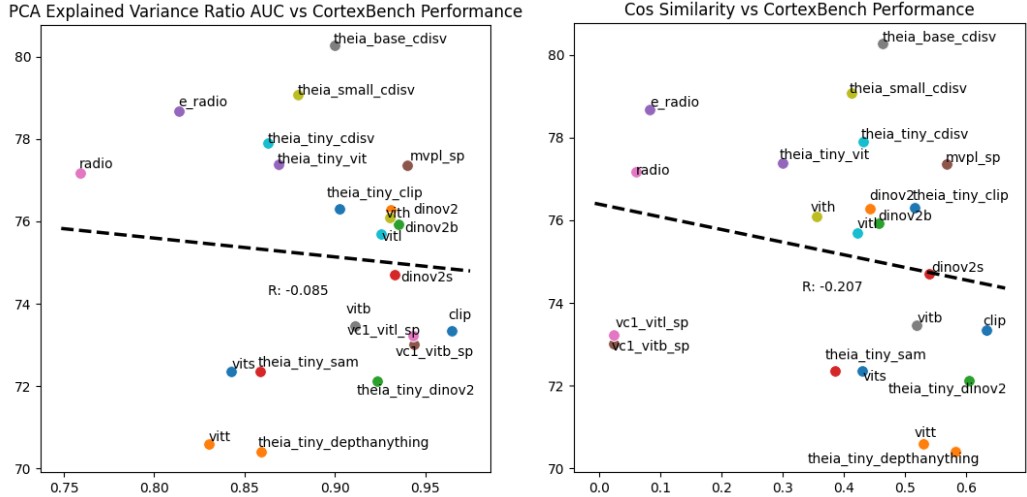

Figure 16: PCA explained variance ratio-AUC (left) and cosine similarity (right) vs MuJoCo performance of many models evaluated.

Table 18: Generalization evaluation on Factor-World.

| Model | Light | Table Texture | Table Pose | Camera Pose | Floor Texture | Arm Pose |
|---|---|---|---|---|---|---|
| Theia-B | 100% | 90% | 100% | 90% | 100% | 55% |
| E-RADIO | 100% | 85% | 80% | 90% | 100% | 40% |
| MVP-L | 100% | 50% | 20% | 70% | 100% | 45% |
| R3M | 100% | 35% | 70% | 90% | 100% | 50% |

