# OpenReview forum: "Theia: Distilling Diverse Vision Foundation Models for Robot Learning"
_robot-learning.org/CoRL/2024/Conference — CoRL 2024_

### Official Review · Reviewer_GPK7 · 2024-07-07
**Ensemble Distillation Method of VFMs**

**Originality:** 1
**Technical Quality:** 3
**Clarity Of Presentation:** 2
**Potential Impact:** 2
**Recommendation:** 2
**Confidence:** 3

**Review:**

This paper distills multiple VFMs from CLIP, SAM, DINO, Depth-Anything, ViT for robotic tasks, similar to RADIO. Compared to RADIO, the main differences are the introduction of new VFMs, and the use of CLS tokens (which are minor). On simulation and real robotic tasks, Theia shows competitive performance and efficiency in the inference.

In order to learn from multiple teachers, it applies a feature translator to match the teacher representation and uses cos-L1 loss for the distillation, following RADIO.

The distilled model, called Theia, is shown to scale effectively, achieving good performance with less MACs. Moreover, the authors did some analysis on the use of CLS tokens and visualization of the ensemble representation. Finally, the authors show that the entropy of feature norms might be related to the quality of the visual representations.

Overall this paper is an extension of RADIO. The distillation method has the same cos-L1 loss and the same hyperparameters. The PCA visualization is also the same. Novelties include the removal of CLS token and the analysis of feature norm entropy. However, I'm not sure if this is enough for the acceptance in CoRL.

**Quality Of The Limitations Section:**

2

**Questions For Rebuttal:**

1. Line 131-132: how are the teacher representations normalized? Is it batch normalization or layer normalization?
2. Could you explain more on the spatial tokens?
3. Why does the CLS token encodes less information than the spatial tokens?
4. Line 216: as far as I understand ViT is more like an architecture than a special model.
5. The formula for the entropy of the feature norms needs to be written down.

**Robotics Focus:**

4

**Summary Of Paper:**

This paper produces a new VFM that distills knowledge from multiple existing VFMs such as CLIP, SAM, DINO, etc.

**Summary Of Recommendation:**

Overall the novelty against RADIO is limited.

---

### Official Review · Reviewer_TPg9 · 2024-07-20
**Review for Theia: Distilling Diverse Vision Foundation Models for Robot Learning**

**Originality:** 3
**Technical Quality:** 4
**Clarity Of Presentation:** 4
**Potential Impact:** 3
**Recommendation:** 3
**Confidence:** 4

**Review:**

Strengths:
- The manuscript studies a very interesting and timely problem.
- The distilled model Theia has been extensively evaluated across simulation and real-world environments, and achieves state-of-the-art accuracy outperforming all examined baselines.
- The proposed model also achieves a significant improvement in inference efficiency, offering an improved speed-accuracy trade-off that is particularly relevant for resource-constrained robot platforms.

Comments:
- Although the results demonstrate the effectiveness of the proposed approach, and thus its contribution, the novelty of this submission is more limited. As the manuscript already discusses, a similar distillation has been proposed in [50] and the differences outlined in Sec. 2.3 can be considered an extension/ablation study of this work. Similarly, the discussion on the feature norm entropy is also very interesting, but largely inspired by the work of [24], applied on a different task.
- The contribution of the paper could also be improved if the naive equally-weighted distillation scheme across models, was replaced by some more advanced or adaptive weighting scheme, e.g. taking into consideration the loss magnitude or the insights provided by the feature norm.
- Some of the arguably unexpected results that arose during the ablation study (e.g. lack of effectiveness from distilling SAM features) have not been further analysed to provide more insights on the underling reasons.

Presentation:
- The paper is really well-written and easy to follow, adequately describes related work and background information, and features
- The accompanying video is also helpful, and demonstrates the conducted real-world experiments and qualitative results.

**Quality Of The Limitations Section:**

1

**Questions For Rebuttal:**

- Do the authors have any insights, or qualitative observations, that justify the poor performance of Segment-Anything and Depth-Anything models? Such visual cues can be considered very important for robot manipulation, and it is surprising that their participation in the proposed distillation scheme downgrades accuracy.

- Would a more naïve fusion method (e.g. feature vector averaging or appending) on the distillation of different models be comparably effective with the proposed feature translators? Experiments indicated that this is not a good option for off-the-shelve VFM fusion, but under the finetuning scheme of the proposed distillation approach, the contribution of the feature translators remains unclear.

- It is unclear whether the authors intend to open-source the weights of the proposed Theia model. Since the SoTA accuracy of the model is in the reviewers opinion the most important contribution of this submission, sharing it with the community would enhance the argument for paper acceptance.

- How could the insights obtained by the proposed Feature Norm entropy metric be fed back to the training process, to improve its accuracy or data efficiency ?

**Robotics Focus:**

4

**Summary Of Paper:**

This submission introduces Theia, a vision foundation model for robot learning applications, distilled for multiple foundation models for primitive vision tasks. During training Imagenet samples are processed through several VFMs (Clip, Dino,ViT etc) and their resulting spatial features are used for distillation-based training of another ViT-based transformer encoder, exploiting some learnable feature translator modules, to conclude to a unified representation. Additionally, an interesting insight that correlates the entropy of features norms with its effectiveness to robot learning tasks.

**Summary Of Recommendation:**

Overall, the manuscript contributes a new state-of-the-art vision foundation model for robotics, achieving state of the art accuracy on robot learning tasks. The paper is well-written and the results are convincing. The proposed distillation scheme and visual representation evaluation approach, although particularly effective, can be considered incremental to existing approaches.

---

### Official Review · Reviewer_CYjq · 2024-07-25

**Originality:** 1
**Technical Quality:** 2
**Clarity Of Presentation:** 4
**Potential Impact:** 2
**Recommendation:** 3
**Confidence:** 4

**Review:**

This is a straightforward paper that leverages existing knowledge distillation techniques to learn a “multi-purpose” pretrained visual representation that captures the different inductive biases of existing models such as CLIP, DINOv2, SAM, and ViTs pretrained for ImageNet classification; the motivation of this work is that each of these features captures something different, and agglomerating/distilling them into a single model is strictly beneficial for a host of robotics tasks.

The evaluations on various forms of policy learning (via behavioral cloning or reinforcement learning) on top of the learned representations capture several different simulation environments and real-world environments, with different robot embodiments and task objectives. In each of these settings, the results show marginal improvement (in many cases, the gains are not significant) over baselines spanning individual representation backbones, as well as what I’d consider a compelling set of “strong baselines” involving channel-wise stacking of existing backbones, evaluating different feature variants from each model (CLS token or spatial tokens), as well as model scales. Finally, the analysis of the internals of the various representations are straightforward (scaling, register features, etc.).

Unfortunately, there are two weaknesses that I feel need to be addressed:

W1. **Evaluation Strategy & Practical Implications** — All representations in this work are fine-tuned for single task policy learning, and evaluated in-distribution (i.e., evaluation initial conditions like object positions, lighting, etc. are sampled from the same distribution as training). This does not feel sufficient, especially given that in many of the experiments, the gains over baselines are marginal. Following the prior work in evaluating representations for robotics, I think this paper would strictly benefit from targeted robustness and generalization evaluations — for example, those in Xie. et. al (https://arxiv.org/abs/2307.03659).

Furthermore, the decision on when to freeze/finetune through the visual representation backbone is not clear, and only on looking at the Appendix did I realize that in some cases, frozen representations fail completely (0% success rate on Toy-Microwave). Why is this the case? If fine-tuning through the backbone strictly helps (and seems more in line with what existing policy learning methods like Diffusion Policy, ACT, Behavior Transformers, etc. do), why not evaluate this directly? Given the large body of work in robot learning and pretrained visual representations, it feels like the time for evaluating frozen visual representations in small-scale, individual target task finetuning experiments is over — I really think we need to look to the complex, multi-task settings, with clear robustness and generalization evaluations. Anything less fails to meet the way existing representations are used practically in existing policy learning pipelines, and seriously hurts the impact of work like this!

W2. **Failure to Evaluate “Diversity” of Visual Representations** — A claim made through the beginning of the paper is the importance of combining diverse pretrained features (and the different inductive bias that each of them provide) for improved downstream performance across robot learning tasks. Unfortunately, the rest of the paper only focuses on rather simple, single-task policy learning as the only evaluation task. Robot learning spans more than just this type of policy learning, compressing visual features to inform action prediction — for example, policies that output keypoints such as PerAct/VoxPoser, or many of the policies trained for longer-horizon manipulation (see papers that evaluate on RLBench as an example). Similarly, other, more modular policies output bounding boxes, operate over multiple image frames, and encode temporal history. Compressing robot learning to just “visual features → simple policy head → action prediction” seems like it’s missing a lot of the power of the Theia representations... I would highly encourage the authors to expand the set of evaluation tasks to optimize for this diversity, and actually estimate things like the representation’s ability to understand low-level shape priors as well as higher-level semantic priors.

**Quality Of The Limitations Section:**

1

**Questions For Rebuttal:**

Q1. Given that you’re learning the feature translators anyway, why is the feature normalization step (Section 3.2) necessary? Doesn’t this by nature change the feature norm distribution across patches (nullifying the analysis at the end of the paper)?

Q2. The paper is currently missing a lot of important details about the evaluation. While many of the task settings are from the VC-1 paper, it would be worth going into more detail about the exact setup here — especially since in the Appendix it seems that some details of the VC-1 evaluation were tweaked for this work. For the real world experiments, how many demonstrations were collected, and with what interface? Who collected them? What does the initial state distribution actually look like? How did you choose the policy architecture? What hyperparameters did you sweep over?

**Robotics Focus:**

4

**Summary Of Paper:**

This work presents a simple approach for distilling existing pretrained visual representation models such as CLIP, DINOv2, and SAM into a unified, smaller model. Through straightforward application of knowledge distillation techniques, the resulting family of Theia models (at different parameter counts) show marginally stronger results on policy learning (behavioral cloning and RL) in both simulation and the real-world compared to individual representations and naive agglomeration strategies.

**Summary Of Recommendation:**

Based on the above weaknesses and lack of important experimental details, I recommend rejection at this time.

---

### Author Rebuttal · Authors · 2024-08-09

**Revised paper is available**

We have revised our paper to reflect our efforts on additional evaluations and clarifications. The revised paper is available in the attached file and the modified parts are highlighted in **green**.

---

### Decision · Program_Chairs · 2024-09-04

**Decision:**

Accept

**Comment:**

This paper presents Theia, a distilled vision foundation model for robot learning, combining features from multiple pre-trained models. The reviewers acknowledge the paper's timeliness and extensive evaluation across simulated and real-world environments, showing competitive performance and improved inference efficiency. However, concerns are raised about the novelty of the approach, given its similarity to previous work like RADIO. The evaluation strategy is questioned, with suggestions for more robust generalization tests and diverse robot learning tasks beyond simple policy learning. While the feature norm entropy analysis is interesting, reviewers note it could be further developed.
Following the rebuttal, two of the three reviewers engaged in discussions with the authors, and the majority of their concerns were addressed, with both leaning towards accepting the paper.